# The Glioma Immune Landscape: A Double-Edged Sword for Treatment Regimens

**DOI:** 10.3390/cancers15072024

**Published:** 2023-03-28

**Authors:** Sukrit Mahajan, Mirko H. H. Schmidt, Ulrike Schumann

**Affiliations:** Institute of Anatomy, Medical Faculty Carl Gustav Carus, Technische Universität Dresden School of Medicine, Fetscherstr 74, 01307 Dresden, Germany

**Keywords:** glioma, immune landscape, glioblastoma therapy, immunotherapy, clinical trials

## Abstract

**Simple Summary:**

Glioblastoma (GBM) is the most severe and aggressive form of primary brain tumor with a poor prognosis. Currently, the treatment for GBM treatment involves surgical resection, followed by radiation and chemotherapy. However, these treatments have shown little success against the disease, with patients having 15–18 months of median survival post diagnosis and a 5-year survival rate of less than 5%. In recent times, scientists have identified potential targets for treating GBM using immunotherapy. However, even using an immunotherapeutic approach has its own challenges in treating GBM. Therefore, for modulating immune cell populations to counter GBM cells, it is essential to expand our knowledge of their role within the tumor microenvironment. Thus, this review will focus on the role of different immune cell populations found in the GBM microenvironment and how they can be modulated for eliciting an efficient immune response against GBM.

**Abstract:**

Immune cells constitute a major part of the tumor microenvironment, thereby playing an important role in regulating tumor development. They interact with tumor cells, resulting in the suppression or promotion of glioma development. Therefore, in recent years, scientists have focused on immunotherapy that involves enhancing the immune response to fight the battle against cancer more effectively. While it has shown success against different cancer types, immunotherapy faces major roadblocks in glioma treatment. These involve the blood brain barrier, tumor heterogeneity and an immunosuppressive glioma microenvironment, among other factors. Additionally, the interaction of the peripheral immune system with the central nervous system provides another challenge for immunotherapeutic regimens. For modulating different immune cell populations to counter glioma cells, it is important to expand our knowledge about their role within the glioma microenvironment; therefore, herein, we review the different immune cell populations found in the glioma microenvironment and navigate through the various shortcomings of current immunotherapies for glioma. We conclude by providing an insight into ongoing pre-clinical and clinical trials for glioma therapies.

## 1. Introduction

Gliomas comprise 80 percent of all malignant brain tumors, making them the most common and lethal type of cancer in the central nervous system (CNS) in adults [1,2]. They are classified into astrocytoma, oligodendroglioma and glioblastoma, based on the cell of origin and tumor grade along with other criteria described comprehensively by the World Health Organization [3]. Fifty percent of all newly diagnosed gliomas are classified as glioblastoma (GBM), a highly aggressive form of glioma with a poor prognosis. The current treatment regimen for GBM involves surgical resection, followed by radiation and chemotherapy [4,5,6] and, in recent times, even tumor-treating fields [7]. In everyday clinical practice, GBM is categorized into two distinct forms: newly diagnosed GBM (ndGBM) and recurrent GBM (rGBM). ndGBM is usually treated with temozolomide (TMZ), an alkylating drug that triggers DNA damage in cancer cells [8]. rGBM treatment involves the use of an inhibitor against vascular endothelial growth factor-A (VEGF-A) called bevacizumab [9], which inhibits angiogenesis within the TME. However, these treatments have shown little success against the disease, with patients having 15–18 months of median survival post diagnosis and a 5-year survival rate of less than 5%. More recently, tumor-treating fields have provided some success along with chemotherapeutic treatment, which involves using alternating electric fields of low intensity (1–3 V/cm) and intermediate frequency (~100–500 kHz) that disrupt cell division of glioma cells [10]. Nonetheless, in most cases, the patients succumb to death due to tumor recurrence.

Various factors are responsible for treatment regimens showing little success against glioma. The presence of the blood–brain barrier (BBB) serves as the foremost obstacle preventing the passage of drugs to the tumor site, thereby restricting appropriate therapeutic intervention. It is constituted by the brain microvascular endothelial cells and pericytes in conjunction with resident astrocytes that form tight junctions and regulate molecular and cellular movement from the blood to the brain. However, in glioma the BBB has been found to be heterogeneously disrupted [11,12] but is still sufficiently intact to limit the optimum delivery of drugs to the tumor site.

Other than BBB being heterogeneous, the cellular composition of the glioma microenvironment (GME) is responsible for providing an extra layer of complexity. The diversity of cells (ranging from brain-tumor propagating cells (BTPCs) to differentiated glioma cells, along with endothelial cells and pericytes constituting the tumor blood vessels, resident neurons and most importantly the immune cell population) affects tumor progression. In addition to cellular heterogeneity, molecular heterogeneity within glioma cells also exists in the tumor microenvironment; this has been reviewed elsewhere [13,14]. In the last decade, finding possible ways to target BTPCs has been a popular idea to counter tumor recurrence. Despite surgical resection along with radiation concurrently with temozolomide [15], BTPCs survive and are regarded as the main drivers of recurrence and tumor progression in glioblastoma due to their ability of treatment resistance and avoiding cell death.

Apart from these factors, the GME plays an important role in tumor progression due to its immunosuppressive nature. Glioma cells express various immunosuppressive factors such as programmed cell death 1 ligand (PD-L1), which restricts tumor antigen presentation [16,17]. In addition, the glioma cells also secrete cytokines and chemokines that attract immunosuppressive components within the GME. Other than glioma cells, the GME hosts various immune cell populations, including tumor-associated macrophages (TAMs), resident microglia, myeloid derived suppressor cells (MDSCs), T cells, natural killer cells (NK cells) and extremely few B-cells, among other immune cells (Figure 1). TAMs and resident microglia constitute approximately 30% of the cellular composition of the GME [18]. The TAMs possess a range of phenotypes from a pro-inflammatory M1 state to an anti-inflammatory M2 phenotype [19]. Human gliomas induce a shift in the polarization of TAMs from an M1 phenotype to an M2 phenotype, thereby suppressing the local immune reaction [20,21]. Thus, in recent years, various studies have implicated TAMs in promoting tumor cell proliferation along with the induction of an immunosuppressive environment by attracting T-regulatory cells (T-regs) and myeloid derived suppressor cells [22,23,24]. In addition to these cells, MDSCs maintain an immunosuppressive environment by inhibiting various effector cells and promoting T-reg function [25,26]. Therefore, factors derived from glioma cells reprogram immune cell populations and provide an environment for the optimum growth and progression of glioma cells.

Thus, it has become increasingly important to develop immunotherapies that can counter glioma-associated immunosuppression and curb glioma progression. For this reason, it is important to gain a better understanding of the glioma immune landscape. In this review, we argue that the immune cell population in the GME serves as a double-edged sword. On the one hand, they are recruited by glioblastoma cells and induce an immunosuppressive environment within the tumor. However, on the other hand, by being modulated in an immunotherapeutic manner, the same immune cells can be used to fight against the very cells that recruited them in the first place. In this context, we review the role of different immune cell populations within the GME followed by discussing the shortcomings of already developed immunotherapies against glioma. Finally, we conclude by providing an insight into the ongoing pre-clinical and clinical trials for drugs in the field of immunotherapy that specifically exploit the immune cells.

## 2. Glioma Immune Landscape

### 2.1. Microglia and Macrophages

Microglia are true CNS parenchymal macrophages and constitute 5–10% of total brain cells. They were initially believed to originate from the neuroectoderm; however, this notion was superseded when Ginhoux et al. showed that microglia arise from embryonic yolk sac (YS) precursors [27,28]. Besides microglia, the YS precursors also give rise to the tissue macrophages. There are three different types of macrophages found in the CNS, namely meningeal, perivascular and choroid plexus macrophages. All three types of macrophages are embryonic in origin; however, the choroid plexus macrophages also arise from adult hematopoietic cells [29]. Together, CNS macrophages and microglia constitute the innate immunity of the CNS and are responsible for the maintenance of normal brain functioning and homeostasis, therefore their role in curbing glioma growth seems indispensable.

Tumor-associated macrophages and microglia constitute 30% of the cellular architecture in GME. Macrophages exhibit high functional plasticity and have been shown to alter their cell surface marker expression. It is now widely established that these cells express M1 (pro-inflammatory) and M2 (pro-tumorigenic) phenotypic markers [30]. The pro-inflammatory M1 macrophages exert anti-tumorigenic effects, while the pro-tumorigenic M2 macrophages encourage tumor invasion, enhance angiogenesis and suppress the immune response [31,32,33]. The M2 state is further categorized into M2a, M2b, M2c and M2d. The different subtypes have been established based on their marker expressions and cytokine secretion profiles; they have been discussed elsewhere [34,35,36]. Similarly, recently, several studies have shown that microglia also possess the potential to polarize into M1- or M2- phenotypes [37,38,39,40]. The M1 phenotype is driven due to tumor necrosis factor α (TNF-α), interferon gamma (IFN-γ) and lipopolysaccharide (LPS), while the M2 phenotype is induced by interleukins -4, -10 and -13 (IL-4, IL-10 and IL-13) [41] (Figure 2). Furthermore, studies have shown that surface markers such as CD80, CD86, human leukocyte antigen-DR isotype (HLA-DR) and CD197 are elevated in M1-macrophages, while M2 macrophages overexpress arginase-1 (Arg-1), CD206, CD163 and CD204 [42,43,44].

Several researchers have observed an increase in M2 macrophages, marked by CD68 and CD163 expression, in high-grade gliomas. Concurrently, a lower number of M1 macrophages are present in high-grade gliomas, as evidenced by the downregulation of the chemokine ligand 3 (CCL3) marker [22]. Additionally, glioma-derived macrophage/microglia cytokine colony-stimulating factor-1 (CSF-1) induces a shift towards the M2 phenotype in macrophages and microglia [45]. This results in the induction of an immunosuppressive environment within the GME due to the release of IL-10 [46]. Moreover, a recent study by Azambuja et al. has shown that GBM-derived exosomes (GBex) convert anti-tumoral macrophages into pro-tumoral macrophages. This leads to the newly reprogrammed M2 macrophages to produce Arg1+ exosomes, which induce an immunosuppressive environment and promote tumor progression [47]. While BTPCs secrete CSF-1 to reprogram macrophages, the inhibition of CSF-1 [45] shifts the M2 macrophages to the M1 phenotype. Furthermore, the silencing of upregulated astrocyte elevated-gene 1 leads to a decrease in the M2-polarization of microglia and an elevation of TMZ-induced DNA damage in glioma cells [48]. Although TAMs support tumor growth and progression, it would be important to see whether they can be reprogrammed to inhibit tumor growth, thereby serving as a double-edged sword in our fight against glioma.

Most of the studies describing the distinct functional phenotypes of the macrophages are based on in vitro experiments using canonical chemokines for inducing polarization. However, in vivo experiments portray a more complex picture of these phenotypes. Recently, immunohistochemistry and single-cell RNA sequencing of tumor samples from humans and animals revealed that different polarization states coexist and, upon analysis of single TAMs, the two phenotypic states were detected at the same time in the same cell [43]. Thus, these phenotypic states do not exist as distinct activation states, as was thought previously, but rather exist as a continuum in macrophages [19,49,50].

On similar lines, a transcriptomic analysis of human macrophages uncovered additional activation pathways outside the standard M1/M2 polarization paradigm that respond to other cytokines and drugs [51,52]. Moreover, other stimuli such as hypoxia and metabolic intake have also been linked to macrophage polarization. While M1 macrophages are normally found in normoxic tumor regions, the M2 macrophages are present in hypoxic conditions of the tumor [53,54]. The discovery of new stimuli and their effect on macrophage polarization suggests the need for using a combination of markers for defining a specific activation state of macrophages, rather than using the nomenclature of M1/M2 phenotypes.

From the studies discussed so far, there are clear indications that targeting molecules that convert M2 to M1 macrophages could serve as a potential therapeutic strategy against glioma. However, recent evidence showing the plasticity of macrophages and microglia serves as a warning here. Instead of focusing exclusively on individual molecules, specific pathways or mechanisms that are employed in the polarization of TAMs need to be targeted.

### 2.2. Myeloid-Derived Suppressor Cells (MDSCs)

Myeloid-derived suppressor cells (MDSCs), as the name suggests, are a subset of myeloid cells known to provide a conducive environment for tumor growth through their immunosuppressive properties. These cells have been shown to be elevated in glioma, besides other cancer types such as melanoma, gastric, endometrial, renal and pancreatic cancer [55,56,57]. Raychaudhuri et al. identified 5.4% of total cells in human GBM as MDSCs and 8% in murine GBM [58]. Functionally, MDSCs constitute neutrophil and monocytic populations and have been categorized broadly into granulocytic-MDSCs (G-MDSCs) and monocytic-MDSCs (M-MDSCs). The G-MDSCs suppress immune cells through the production of arginase-1, reactive oxygen species (ROS) and prostaglandin E2, while M-MDSCs express arginase-1 and PD-L1 and secrete IL-10 and TGF-β for inducing an immunosuppressive environment [59,60,61,62,63] (Figure 2). In addition, IFN-γ secreted by cytotoxic cells against tumor cells has also been shown to upregulate PD-L1 expression in MDSCs within the GME, thus encouraging an immunosuppressive state of GBM [58]. Bayik et al. observed that the infiltration of G-MDSCs vs. M-MDSCs in mouse models of GBM was sex-dependent. While male mice showcased an enrichment of M-MDSCs, female mice had elevated levels of G-MDSCs in their blood samples [64], which, on being targeted in respective sexes, led to improved survival. Similarly, MDSCs were shown to be significantly increased in blood samples of GBM patients; however, patients with lower grade glioma showed only a slight and non-significant increase.

Other than G-MDSCs and M-MDSCs, a set of early MDSCs (eMDSCs) have also been identified in the GME that do not express distinct surface markers from either set of mature MDSCs. Mi et al. clearly described M-MDSCs with CD11b+ CD14+ CD33+ HLA-DRlow/- CD15- signatures, while G-MDSCs were shown to contain CD11b+ CD14- CD33+ HLA-DRlow/- CD15+ signatures on their surface in humans [26]. The eMDSCs are CD45+ CD3- CD14- CD15- CD19- CD56- HLA-DR- CD33+ CD13+, thus clearly separating the eMDSC population from the other two [65]. The authors also define the signatures of these cells in mice, with M-MDSCs expressing CD11b and Ly6C, while G-MDSCs express CD11b and Ly6G (CD11b+ Ly6G+ Ly6C-). eMDSCs are characterized in mouse as CD11b+ Ly6C- Ly6G- F4/80- MHCII-. The eMDSCs have previously been identified in various other cancer types, including head and neck cancer and ovarian cancer, although their numbers were equivalent to those found in healthy blood samples [66,67]. Studies have now shown that eMDSCs are elevated in the blood samples of glioma patients compared with healthy donors. However, a recent study found a substantial overlap of markers between basophils and eMDSCs [68] and indicated that most of the population previously thought of as eMDSCs could possibly have basophil contamination due to the overlap. Therefore, more investigations need to be undertaken for confirming the presence of eMDSCs in glioma patients and for determining their role in glioma development.

In the case of G-MDSCs and M-MDSCs, contrasting reports regarding their presence in glioma patients have been published in recent times. While Raychaudhuri et al. found elevated levels of G-MDSCs in the blood samples of GBM patients [58], Dubinski et al. described M-MDSCs as the most prevalent population, amongst the MDSCs, in the peripheral blood samples of GBM patients [69]. Interestingly, in the glioma tissue, Raychaudhuri et al. found that eMDSCs comprised the majority of MDSCs, followed by G-MDSC (CD14- CD15+) and M-MDSC (CD14+ CD15-) subtypes. Dubinski et al. also reported elevated levels of G-MDSCs compared with M-MDSCs in the glioma tissue [69]. Further, Gielen et al. showed that MDSCs found in the tumor tissue of GBM patients are mostly granulocytic in nature [70]. However, Alban et al. identified M-MDSCs forming a majority in the GME compared with other MDSCs [71]. These studies clearly indicate the existence of different levels of MDSC subtypes in the peripheral blood and glioma mass of patients, although G-MDSCs are being suggested to be the major MDSC subtype within the glioma mass.

Several studies, as mentioned above, have now showcased that MDSCs use a wide variety of mechanisms to suppress a cytotoxic immune response in GBM, making them solid targets for glioma therapy (Table 1). In one of these studies, in murine GBM tumors, a reduced infiltration of G- and M-MDSCs was observed during treatment with the tyrosine kinase inhibitor sunitinib. In addition, an increase in CD3+ and CD4+ T cells was observed [58]. In another study, a macrophage migration inhibitory factor (MIF) was identified that enhances the immunosuppressive capacity of MDSCs [72] in late-stage melanoma patients. In addition, it has been linked to breast and lung cancer [73,74]. Otvos et al. identified the same factor being secreted by GBM cells and stimulating MDSC function in glioma [75]. In fact, MIF expression has been correlated with the severity of the disease, with high levels of MIF resulting in poor prognoses for glioma patients. According to a model suggested by Alban et al., MIF secreted by GBM cells binds to CD74 present on MDSCs, which propels them to inhibit CD8+ T cell function [71]. Highly conserved in mammals, MIF is also produced by various immune cell populations, including T cells, macrophages, monocytes and neutrophils, and its function as part of tumor immunity have been well described elsewhere [76]. Other than proteins such as MIF, glioma cells release certain RNA molecules such as microRNA-29a (miR-29a) and microRNA-92a (miR-92a), which have been implicated in the modulation of MDSC function. miR-29a silences high-mobility group box transcription factor 1 (Hbp1), resulting in cell cycle progression and differentiation of MDSCs to an immunosuppressive state. Similarly, miR-92a suppresses the expression of protein kinase cAMP-dependent type I regulatory subunit alpha (Prkar1a) present on MDSCs [77]. The silencing of Prkar1a results in the upregulation of immunosuppressive factors TGF-β and IL-10 that support a favorable environment for glioma progression (Figure 2).

Over the years, various molecules have been identified that drive MDSC function within glioma [84,85,86]. However, our knowledge regarding how most of these molecules mechanistically modulate MDSC function and ways to target them is still lacking. Many questions remain unanswered, including the infiltration pattern of these cells within the tumor tissue and whether one type is more favored than others by the glioma cells. Moreover, it is unclear whether MDSCs, like macrophages and microglia, can switch between different subtypes due to environmental cues or whether these are discrete activation states, as already described elsewhere [87].

### 2.3. T Lymphocytes

In glioma, the knowledge about the functional state of the infiltrating T cell population is very limited. Patients suffering from glioblastoma showcase disrupted BBB which in turn leads to an increased infiltration of T cells in the GME. However, T cells fail to induce an attack on glioblastoma cells since the latter downregulate major histocompatibility complex (MHC) expression, thereby preventing antigen presentation and avoiding recognition by T cells [88]. Moreover, the glioma cells exhibit an elevated expression of programmed death ligand-1 (PD-L1), which binds to the PD-1 receptor on T cells and inhibits T cell proliferation, cytokine production and cytolytic function [78]. It is sufficient to say that GBM causes severe T cell dysfunction; Grabowski et al., along with others, have molecularly categorized this dysfunctional state of T cells in glioblastoma into five domains, namely senescence, tolerance, anergy, exhaustion and ignorance [89,90,91].

Along with TAMs, regulatory T cells (Tregs) are responsible for providing an immunosuppressive environment for glioma cells (Figure 3). Tregs constitute a unique subpopulation of helper T cells through the expression of the FoxP3 transcription factor, which is required for the induction of immunosuppression [92]. The Treg population in human glioblastoma samples is increased fourfold compared with benign adenomas and meningiomas [93,94]. These cells promote tolerance of glioma cells by other effector T cell populations (CD4+ and CD8+) via secretion of transforming growth factor β (TGF-β) and IL-10. This results in the depletion of IL-2 and IFN-γ, which are needed for the development of cytolytic T cells [95]. The cytolytic T cells are known to induce apoptosis in tumor cells via the FasR-FasL pathway and through secretory granules containing granulysin, perforin and granzymes [96]. However, cytolytic T cell action is inhibited due to the immunosuppressive environment induced by Tregs.

Recently, a subset of Tregs has also been characterized into T follicular regulatory (Tfr) cells that express FoxP3 along with the Bcl-6 transcription factor. These cells recognize tumor neo-antigens and, on encounter with an antigen, undergo clonal expansion. Eschweiler et al. reported the prevalence of these cells in tumor tissues of different cancer types and demonstrated the differentiating aspects of Tfr cells from Treg cells [79]. Single cell RNA-sequencing results showed two distinct CD4+ T cell clusters, enriched for FoxP3 expression. These clusters also exhibited very distinct transcriptomic signatures and the Tfr cells expressed high levels of immunosuppressive proteins, such as cytotoxic T lymphocyte-associated antigen-4 (CTLA-4), IL-10 and TGF-β1, compared with the Treg cells. The authors concluded that, although Treg and Tfr cells share clonotypes, the Tfr cells showed more clonal expansion compared with the Treg cells [79]. Other than FoxP3 and Bcl-6, these cells also expressed programmed cell death protein 1(PD-1) and CTLA-4. Interestingly, in syngeneic mice models for melanoma, treatment with anti-PD-1 led to an increase in Tfr cells within the tumor microenvironment, which was detrimental for mice survival. However, Tfr depletion with anti-CTLA-4 followed by anti-PD1 treatment improved survival in mice, indicating that monotherapy with anti-PD-1 or anti-CTLA-4 would not be beneficial [79]. Until recently, there was very little known about the role of Tfr cells in glioma. Lu et al. found increased levels of these Tfr cells, along with Treg cells, in resected glioma samples, which significantly suppressed CD8+ T cell proliferation and cytotoxic capacity [80]. In fact, these cells demonstrated a greater potency for suppression in glioma tumor samples compared with blood samples obtained from healthy subjects and patients suffering from glioma. These studies provide an insight into the potential that Treg and Tfr cells hold as targets for glioma therapy (Table 1), although many open questions remain regarding glioma development with T cells as the vantage point.

### 2.4. B Lymphocytes

B cells secrete immunoglobulins and constitute the humoral immunity. They suppress tumor growth by the promotion of a T cell response [97]. However, very little is known about the role of B cells in glioma. Recent evidence suggests a role for a subpopulation of B cells, called regulatory B cells (Bregs), in glioma development. Essentially, Bregs induce an immunosuppressive environment similar to Tregs by producing IL-10, IL-35 and TGF-β [98].

In recent years, there have been very few reports describing the role of these cells in the GME. Han and colleagues showed in 2014 that the glioma cell-derived placental growth factor (PGF) led to the differentiation of tumor infiltrating B cells into Bregs by inducing TGF-β expression. Further, the authors observed that Bregs suppressed CD8+ T cell response in vitro [81]. Although more investigations are required for deciphering the role of Bregs, their possible presence in the GME could add to the immunosuppressive phenotype induced by microglia, macrophages and Tregs.

### 2.5. Natural Killer (NK) Cells

Natural killer (NK) cells share a common progenitor with B cells and T cells but belong to the innate immune system, in comparison to the latter two, which are part of the adaptive immune response. While T cells require priming by antigen presenting cells, NK cells are activated in response to interferons or cytokines secreted in their vicinity. Accordingly, NK cells can kill infectious viral particles and, more importantly, cancer cells without any priming [99]. The NK cells contain granules rich in perforin and granzyme, which, on being released in the vicinity of a target cell, create pores in the cells through which granzyme enters and induces apoptosis.

In glioma, NK cells constitute the least abundant population amongst the immune cells that infiltrate the tumor mass. The cells carry out their function by interacting with tumor cells through a combination of stimulatory and inhibitory receptors [100]. For example, NK cells have killer cell immunoglobulin-like receptors (KIR) that induce an inhibitory effect on NK cells upon recognizing MHC class I molecules on host cells. Thus, despite downregulating the MHC class-I expression to evade detection by T cells, glioma cells can still be targeted by NK cells [94,101]. Several studies have already established the importance of NK cells in cancer research by highlighting how NK cell deficiency leads to a greater risk of developing cancer [102,103,104,105]. Incidentally, these cells have not been as well explored as T cells for treating glioblastoma [106]. Interestingly, in GBM patients, NK cells have shown to behave differently compared with healthy patients by expressing low levels of the NKG2D receptor that usually activates the NK cells to carry out cell-mediated killing of the tumor cells [82,107]. In addition, Fu et al. have shown that NK cells express low levels of IFN-γ within the tumor site in glioma [83], further decreasing the interest in exploiting NK cells as a potential glioma therapy. Fortunately, recent studies have made positive strides in exploiting the function of NK cells to curb glioblastoma progression [108,109,110]. For example, in vivo administration of NKG2D-based CAR-T cells (discussed in part 3) led to the prolonged survival of glioma-bearing mice and demonstrated a high production of IFN-γ [108]. While most of these studies are in vitro, they provide substantial evidence that modulating NK cell pathways and activating NK cells holds potential as a glioma therapy (Table 1).

## 3. Currently Known Immunotherapies

We consider the glioma immune landscape as a double-edged sword, serving two purposes: supporting glioma growth but also capable of fighting GBM. In the previous section, we introduced one edge of the sword by describing how immune cells affect tumor growth and, in turn, how glioma cells modulate the function of different immune cell populations to induce immunosuppression. This knowledge motivated the research community to target immunomodulatory molecules and to modulate mechanisms that eventually help in improving the survival of patients suffering from glioblastoma (Table 2). While some immunotherapies aim to enhance the immune system, others directly target the tumor cells. In this section, we shall describe the different types of immunotherapies being used in glioblastoma treatment, followed by their respective limitations. We primarily focus on how different immunotherapeutic approaches affect the glioma immune landscape and in turn sharpen the sword edge fighting against GBM.

### 3.1. CAR-T Cells

Chimeric antigen receptor (CAR) T cell therapy involves genetic engineering of T cells that are isolated from the patient’s blood. The T cells are modified to express chimeric receptors that target tumor antigens and are reintroduced into the patient’s bloodstream. This therapy bypasses the involvement of MHC-dependent antigen presentation, thereby providing an advantage over endogenously circulating T cells. With a great success rate in treating different cancers, including large B cell lymphoma and acute lymphoblastic leukemia, clinical studies are now focused on targeting solid tumors, such as glioblastoma, using CAR-T therapy. Over the years, researchers have identified unique markers in glioblastoma such as the epidermal growth factor receptor variant III (EGFRvIII), human epidermal growth factor receptor 2 (HER2) and IL-13 receptor alpha 2 (IL-13Rα2) as unique targets of CAR-T therapy. Clinical trials for many of these targets have largely been completed (Table 2).

EGFR is amplified, mutated or both in 40% of primary glioblastomas. EGFRvIII, a mutant form of EGFR, occurs in ~60% of EGFR-overexpressing glioblastomas [123,124,125]. EGFRvIII is exclusively expressed by tumor cells and is associated with poor prognoses of patients. EGFRvIII has been shown to be involved in different roles in glioma, including growth of tumor cells, survival, invasion and angiogenesis, among others [125,126]. The different roles of EGFRvIII in glioblastoma have been reviewed elsewhere [123,124,127,128]. IL-13Rα2 is highly expressed in GBM, with an overexpression observed in over 75% of GBMs [129]. Moreover, the absence of IL13Rα2 expression in the brain tissue makes it an important potential target against GBM [130]. Like EGFRvIII, IL-13Rα2 overexpression in glioma is associated with poor patient prognosis; however, its role in glioblastoma has remained controversial. While Brown et al. linked IL-13Rα2 expression to mesenchymal signature gene expression, Tu et al. linked its expression to metastasis and glioma cell growth [131,132]. Newman et al. provided substantial evidence for the role of IL-13Rα2 in triggering invasiveness and promoting GBM cell growth, ending this controversy. The authors observed that IL-13Rα2 alone induces invasiveness of GBM cells and has no effect on proliferation. However, in the presence of EGFRvIII, it promotes the growth of tumor cells [133]. Like EGFRvIII and IL-13Rα2, HER2 protein has been observed to be highly expressed in ndGBM, while secondary GBM showed a low expression of HER2 [134]. Accordingly, CAR-T therapy targeting HER2 underwent a clinical trial and also employed cytomegalovirus (CMV) as a potential target [135]. CMV is found in 90% of cancers, including breast, ovarian, prostrate, colon, sarcomas and GBM [136]. CMV was shown to promote GBM in murine models via pericyte recruitment and angiogenesis [137]. The results from the phase I clinical trial found HER2-CAR-CMV T cells to be safe and clinically beneficial for patients with progressive disease (NCT01109095).

Although clinical trials using CAR-T cell therapy have provided optimistic results for some of the proteins being targeted [138,139,140], the anti-tumor efficacy in these trials was limited due to a highly immunosuppressive GME, tumor heterogeneity and the ability of tumor cells to downregulate unique target antigens [141,142,143]. To overcome these challenges, scientists are engaged in combining CAR-T cell therapy with other therapies and intend to obtain a better synergistic effect.

### 3.2. Tumor Vaccines

Scientists today are targeting different tumors by administering tumor-associated or tumor-specific antigens as bait to elicit an immune response in the patients. These agents are called tumor vaccines. While tumor-associated antigens such as Wilms’ tumor 1 (WT1) protein, heat shock proteins (HSPs) and survivin are overexpressed by tumor cells, they are also found in other cell types. However, tumor-specific antigens such as EGFRvIII are exclusively expressed by the tumor cells and are mutant forms of the wildtype protein.

The identification of antigens specifically expressed in GBM has led to the development of vaccines specific to these antigens. With respect to GBM, scientists have developed a vaccine against the antigen EGFRvIII, called Rindopepimut. While initial clinical trials showed an improvement in patient survival, the ACT IV study was terminated when treatment with Rindopepimut did not improve the overall survival (OS) of patients with ndGBM [112]. In addition, since EGFRvIII is mostly found in ndGBM and rarely in secondary GBM, the use of the vaccine is restricted. Furthermore, EGFRvIII expression is heterogeneous among the tumor cells and, with the added disadvantage of antigen downregulation, Rindopepimut faces major challenges as a treatment option for GBM [144,145]. The Rindopepimut trials have been reviewed extensively elsewhere [145,146].

In recent years, isocitrate dehydrogenase 1 (IDH-1) has garnered interest for treating glioma patients. It is found to be mutated in low grade gliomas, while the wildtype IDH1 protein is mostly associated with primary GBM [147]. The mutation in the protein involves an arginine-to-histidine replacement at position 132 (IDH1R132H). Pre-clinical studies have indicated that vaccines against the mutated protein may elicit an anti-tumor response from CD4+ and CD8+ T cells [148,149]; clinical trials are ongoing. Other than EGFRvIII and IDH1R132H, which are major players in glioma progression and prognosis, vaccines against survivin and WT1 protein have also been developed. Survivin belongs to a family of inhibitors of apoptosis (IAP) proteins and is highly expressed in tumor cells of various cancers, including GBM. It was found to provide radiation resistance to GBM cells by suppressing apoptosis through a caspase-independent pathway [150]. Around the same time, Dohi et al. found that survivin suppresses caspase activity and enhances tumorigenesis in vivo [151], thereby indicating the dual role of survivin in GBM cell survival. Accordingly, the SurVaxM vaccine was developed to target the survivin protein in GBM; early outcomes of the clinical trials indicate high levels of CD8+ T cells along with antibodies against survivin. More importantly, an improvement in OS and progression free survival (PFS) was observed in the Phase II clinical trials comprising ndGBM patients [152]. Like survivin, WT1 protein is found to be greatly expressed in GBM compared with lower grade gliomas and studies have indicated its role in promoting tumorigenesis [153,154,155]. The DSP-7888 vaccine, which was developed against WT1, elicited a T cell response and improved OS in patients. The positive response with DSP-7888 has led scientists to undertake clinical trials involving DSP-7888 with other immunotherapeutic regimens, including bevacizumab or immune checkpoint inhibitors, for targeting ndGBM and rGBM, along with other solid tumors [156,157,158,159].

While some of these peptide vaccines have resulted in improved PFS and OS, they face multiple challenges from the tumor. Peptide vaccines against EGFRvIII, IDH1R132H, survivin and WT1 involve single-antigen targets and this may lead to immune evasion due to the inherent cellular heterogeneity and antigen loss associated with GBM. To counter this, scientists have developed a multi-peptide vaccine that contains 11 glioma-associated antigens, thereby ensuring an onslaught of CD8+ and CD4+ T cell response. This multi-peptide vaccine is known as the heat shock protein (HSP) peptide complex (HSPPC-96) vaccine and consists of a heat-shock protein, glycoprotein 96 (gp96, expressed in glioma) and promotes tumor growth [160,161] attached to autologous tumor-derived peptides. It is currently being tested as a potential therapy against GBM, among other cancers [162,163]. There are several limitations to peptide vaccines other than the challenges introduced by the tumor biology. These vaccines depend on specific antigens expressed in the patient’s tumor and are HLA-subtype specific, as in the case of HSPPC-96. Thus, novel strategies need to be introduced to counter this problem.

Another way to target tumor cells using vaccines involves priming dendritic cells (DCs) derived from the patient with tumor antigens or mRNA-expressing MHC molecules and then administering them back into the patient. Appropriately given the name DC vaccines, these vaccines can elicit an anti-tumor response by activating CD4+ and CD8+ T cells. Currently, a plethora of DC vaccines are undergoing clinical trials, in which DCs have been primed with WT1 protein as well as nucleic acid and protein molecules of cytomegalovirus (CMV). CMV proteins are greatly expressed in 90% of GBMs, and phosphoprotein 65 from the CMV was found to be present in ~70% of these tumors. Due to its presence exclusively in GBM, CMVpp65 serves as a tumor-specific antigen that can be targeted without inducing off-target effects [164]. Other than protein constituents, nucleic acids from CMV have been found in both ndGBM and rGBM [165]. Thus, CMV phosphoprotein 65 RNA (CMV pp65) has also been used for developing DC vaccines. While DC vaccines have shown optimistic results alone, they are also being used in combination with other therapeutic approaches, thus paving the way for combinatorial therapies to become the new norm in our fight against GBM. A detailed review on DC vaccines has been provided elsewhere [166].

### 3.3. Immune Checkpoint Inhibitors

The immune system employs checkpoint proteins for maintaining a balance between tolerating self-antigens and eliminating pathogens. On this basis, the checkpoint proteins are categorized into stimulatory and inhibitory. Tumors exploit these proteins for their benefit, thereby making them associated with tumor growth and progression. Therefore, inhibitors have been developed to target such checkpoint proteins to prevent tumor growth.

In glioma, an upregulated expression of inhibitory checkpoint proteins, PD-1 (and its ligand PD-L1) and CTLA-4, have been associated with immune evasion, increased tumor grade and poor patient prognosis [167].

PD-1 is an immunoglobulin receptor that is expressed by activated T, B, NK and dendritic and myeloid cells. In contrast, the PD-L1 receptor is expressed mostly by antigen presenting cells (APCs) and cancer cells. The PD-L1 and PD-L2 ligands bind to the PD-1 receptor and this interaction negatively regulates T cell population in the GME by suppressing the activation and infiltration of T cells. In addition, it also inhibits the secretion of pro-inflammatory factors such as IFN-γ, thus proving to be pro-tumorigenic in the TME. To counter the effect of PD-1 and its ligands, inhibitors such as nivolumab and pembrolizumab have been developed (Table 2), while many others are undergoing clinical trials. These immune checkpoint inhibitors (ICI) proved to be beneficial in treating melanoma, non-small cell lung cancer (NSCLC) and other solid tumors [168,169,170], however they were largely ineffective in the case of GBM. Although pre-clinical trials showed an increase in CD8+ T cell infiltration and the upregulation of immunologic memory markers on tumor infiltrating lymphocytes (TILs), clinical trials with nivolumab such as CheckMate143 (NCT02017717) [171], CheckMate 498 (NCT02617589) and CheckMate 548 (NCT02667587) [172] failed to meet the desired OS in patients. In contrast, treatment with neoadjuvant pembrolizumab has shown varying results in clinical trials. In one clinical study, a significant increase in the PFS and OS was observed. Further, an upregulation of IFN-γ, augmented T cell receptor clonal diversity in the TILs and infiltration of CD4+ and CD8+ T cells was observed in the clinical trials with pembrolizumab [173]. However, another study employing anti-PD1 monotherapy reported a predominance of immunosuppressive macrophages in GME, even though infiltration of T cells was observed [174]. These studies indicate that mono-therapeutic treatments involving anti-PD1 cannot induce an effective immune response against glioma. In this regard, combinatorial therapies involving pembrolizumab with bevacizumab did not show an improvement in PFS and OS in rGBM patients compared with bevacizumab treatment alone [175].

CTLA-4 is constitutively expressed on Tregs and upregulated in GBM. It binds to the CD80 and CD86 proteins present in DCs, activated B-cells, macrophages and T cells and thereby inhibits their function. In general, CD80 and CD86 bind to the CD28 receptor present in T cells, thereby triggering co-stimulatory signals resulting in T cell activation. However, it would be important to note that CD80 and CD86 have a higher avidity to CTLA-4 than to CD28, thus suggesting a predominance of CTLA-4 inhibitory action versus the co-stimulatory function of CD28 [176]. CTLA-4 expression inhibits IL-2 and INF-γ secretion, thus providing an immunosuppressive environment favorable for GBM cells. Blocking the interaction of CTLA-4 with CD80/CD86 led to tumor shrinkage and improvement in survival in pre-clinical GBM models [177,178]. Interestingly, the CD4+ T cell proliferative capacity was restored exclusively in the CD4+ CD25- population, without affecting the Treg population CD4+ CD25+. In addition, blocking CTLA-4 (using ipilimumab) led to an increased infiltration of CD8+ T cells into the tumor core. The study corroborated its findings by providing evidence that no change in survival was observed in immunocompromised mice on blocking CTLA-4. This clearly indicates that a CTLA-4 blockade stimulated an anti-tumor response that is otherwise suppressed by Tregs [178]. These studies laid the foundation for carrying out clinical studies using ipilimumab. Initial outcomes from clinical studies have provided improvement in survival with ipilimumab alone; however, most trials are employing a combinatorial approach with ipilimumab. Currently, the Ipi-Glio trial involving ipilimumab with temozolomide is being tested to improve the PFS in ndGBM patients [179]. Other than this, VEGF inhibitors and other checkpoint inhibitors are being used along with CTLA-4 inhibitors for deciding the best course of action [180].

Other than PD-1 and CTLA-4, indoleamine 2,3-dioxygenase-1 (IDO-1) as well as T cell immunoglobulin and mucin domain 3 (TIM-3) have recently come to the forefront in glioma immunotherapy. IDO-1 enzyme is involved in catabolizing tryptophan into kynurenine. Under physiological conditions, IDO-1 is expressed by mature DCs and macrophages, as well as being expressed by other cells outside of the immune system. It is activated by IFN-γ, TNF-α and TGF-β among other proteins in disease conditions. The depletion of tryptophan and the production of kynurenine by IDO-1 activates Tregs and MDSCs, thus promoting an immunosuppressive environment [181]. In addition, IDO-1 activity leads to the suppression of effector T and NK cells and the promotion of angiogenesis in solid tumors, while IDO-1 deficiency causes a stark depletion of Tregs and tumor rejection mediated by CD4+ and CD8+ T cells [181]. Interestingly, the peripheral IDO does not directly affect T cell infiltration into the glioma tissue, but cells within the GME expressing IDO-1 drive the intra-tumoral accumulation of Tregs [182]. Moreover, an upregulation of IDO expression in glioma leads to poor patient prognosis. In this regard, a clinical study testing the IDO-1 inhibitor along with nivolumab was undertaken in patients with rGBM, but was terminated (NCT03707457).

TIM-3 is an immunosuppressive receptor protein expressed in CD4+ and CD8+ T cells, Tregs, NK and myeloid cells and has been shown to promote T cell exhaustion [183,184,185,186]. Recently, Guo et al. identified that glioma cell-intrinsic TIM-3 induces macrophage migration and promotes a pro-tumorigenic phenotype by regulating IL-6 expression [187]. While TIM-3 inhibitors have already been tested for treating different cancers such as melanoma, ovarian cancer and gastric cancer, clinical trials against GBM are ongoing [188,189]. In one such trial, the combination of anti-TIM-3 along with anti-PD-1 has been tested in patients with rGBM and the results of the study are eagerly awaited (NCT03961971).

Although checkpoint inhibitors have resulted in positive outcomes in treating GBM, they come with their own set of limitations. An important barrier in this regard is the BBB, which might block the antibody from entering the brain parenchyma. In addition, monotherapies or combinatorial approaches, while successful in pre-clinical studies and early phase trials, have not proven to be optimally efficacious, especially in phase III clinical trials. Therefore, it becomes necessary to rethink current treatment approaches to produce the maximal effect of combinatorial therapies involving immune checkpoint inhibitors.

### 3.4. Oncolytic Viruses

Oncolytic viruses (OVs) selectively lyse tumor cells after replicating within them. Oncolytic virotherapy (OVT) employs native or genetically engineered viruses, which can enhance immunogenicity in the tumor tissue. OVT overcomes the limitation of ICIs by being directly injected intra-tumorally or by using viruses that can penetrate the BBB.

Following the success of OVT in melanoma, nasopharyngeal carcinoma and bladder cancer treatment, it has raised hope in GBM treatment. Interestingly, GBM virotherapy trials already started more than three decades ago and concluded with the selective killing of GBM cells [190]. DNX-2401 (tasadenoturev), a replication competent adenovirus, has a high tumor selectivity since it targets integrins with an RGD motif in GBM cells [191]. In two separate clinical trials of DNX-2401 for rGBM patients, an infiltration of CD8+ and T-bet+ cells (DCs, NK and innate lymphoid cells) was observed in the tumor tissue. Additionally, the cerebrospinal fluid analysis showed elevated levels of cytokines promoting polarization of macrophages to an active pro-inflammatory state (M1 phenotype) [191,192]. DNX-2401 in combination with pembrolizumab is also being tested for rGBM patients [193].

Teserpaturev (G47∆) is an oncolytic herpes simplex virus type 1 (HSV1) being tested in rGBM patients [194]. The biopsy of patients treated with the viral therapy revealed an increased infiltration of CD4+ and CD8+ cells following repetitive injections of the virus [195]. Although the infiltration of immune cells could be a consequence of viral presence, tumor shrinkage was observed 4 months later. Interestingly, FoxP3+ cells were rarely found in biopsy samples; however, tumors that regrew post viral therapy showed an increase in FoxP3+ cells. This clearly implies that inhibiting FoxP3+ cells would be beneficial for glioma therapy with teserpaturev. Finally, a phase II trial with teserpaturev demonstrated a 1-year OS in 84.2% of rGBM patients and has become the first OVT being approved in Japan (UMIN000015995).

Many other oncolytic HSV1 being tested in patients suffering from high-grade gliomas and recurrent gliomas include G207 [120] and rQNestin34.5v.2 (NCT03152318) [196], respectively. rGBM patients that were treated with G207 and radiation demonstrated a T cell and humoral response against the tumor [121] (Table 2). Treatment with rQNestin34.5v.2 led to an increased number of MDSCs in the TME due to CCL2 chemokine released by NOTCH-activated macrophages [197]. Similar results were obtained by analyzing the TCGA database for grade IV glioma that showed significant correlation of higher myeloid infiltration with NOTCH signaling. Other than CCL2, patients treated with rQNestin34.5v.2 showed a presence of IL-10 in their serum samples. Interestingly, the blocking of NOTCH signaling rescued the immunosuppressive phenotype induced within the TME by MDSCs and led to the activation of a CD8+ T cell response [197]. These studies provide significant evidence of the different signaling pathways and mechanisms involved in glioma progression. These in turn can be explored further as potential targets for treating GBM. Oncolytic parvovirus (ParvOryx) was also developed and has been tested in several clinical trials for determining its safety. Results from the study include marked activation of microglia/macrophages and infiltration of cytotoxic T cell populations in infected tumors persisting for several months [198]. Other than DNA-based viruses, different RNA based OVs have also been used in clinical trials, including a chimera of poliovirus and rhinovirus (PVS-RIPO) that showed a 1-year OS in 21% of rGBM patients [119]. OVTs involving other viruses such as Newcastle disease virus and vaccinia virus, among many others have shown promising results in some clinical studies in treating GBM and different glioma types and have been reviewed elsewhere [199,200].

While early phase clinical trials indicate improved survival, the percentage of patients benefitting from OVT have been relatively few [119]. A major hurdle in adopting OVT at the clinical level is its need to be injected intra-tumorally. One way to overcome this challenge is through intravenous (IV) delivery of the OVs. Currently, IV delivery of OVs is still at a nascent stage in clinical trials and faces its own set of challenges, including the optimal dosing of the virus and pre-clearing of the viral particles by the peripheral immune system. However, in this regard, a study involving the IV infusion of oncolytic reovirus against high-grade gliomas showed evidence of an increased infiltration of CTLs and CD68+ macrophages/microglia as well as an upregulation of IFN-regulated gene expression [201]. Very few NK and B-cells were found in the tumor tissue. The study also showed an upregulation of the PD-1/PD-L1 axis in human tumor samples. Interestingly, the authors used a PD-1 blocker along with the reovirus to enhance the immunotherapeutic effect of OVT [201]. These studies encourage us to employ more combinatorial approaches compared with monotherapies, not only for eliciting a better immunogenic response but also for improving the survival in patients suffering with glioma.

## 4. Potential Glioma Immunotherapies

Apart from immunotherapeutic targets discussed in the previous sections, recent reports have uncovered several proteins and pathways that can be targeted for GBM treatment. Targeting some of these proteins shows promising results in pre-clinical trials for glioma therapy and have been described below. Along with new target candidates, we shall also be discussing the challenges in glioma therapy research and ways to overcome them.

In recent years, other potential candidates for glioma therapy have garnered interest. These proteins include, among others, matrix metalloproteinases 2 (MMP2), CD70, CD147, disialoganglioside (GD2), B7-Homolog 3 (B7-H3) and ephrin type-A receptor 2 (EphA2) (Figure 4). Their role in different aspects of tumor progression, including angiogenesis, invasion, metastasis and tumor cell growth, provide us with an insight into different mechanisms that can be targeted to curb GBM progression [202,203,204]. With a priori knowledge that GBM expresses high levels of GD2 protein, Golinelli et al. developed a strategy to deliver a pro-apoptotic factor called TRAIL (tumor necrosis factor-related apoptosis-inducing ligand) in mesenchymal stem cells (MSCs) to GD2-positive tumors. For this, they genetically modified MSCs to express a truncated form of anti-GD2 CAR for selectivity against GD2-expressing GBM models [205]. Marx et al. tested dinutuximab beta, an antibody against GD2, to determine the anti-tumor efficacy of GD2-directed treatment against GBM. This group observed an antibody-dependent cellular cytotoxicity effect against the majority of GBM cell lines [206].

B7-H3 is an immune checkpoint protein that is frequently overexpressed, especially by glioma cells, vessels, tumor-infiltrating DCs and macrophages in the TME of GBM patients. Tang et al. developed a B7-H3 CAR and targeted GBM cells in vitro and in vivo [207]. In vitro, B7-H3 CAR-T therapy led to the lysing of different glioma cell lines such as A172, U87-B7-H3 and UPN1 and the increased secretion of IFN-γ and IL-2 in co-culture analysis. In vivo analysis in a xenograft orthotopic GBM patient-derived xenograft (PDX) mouse model showed complete regression of a tumor that was maintained for about 2 months.

Several in vivo studies and analyses of the TCGA database have established platelet-derived growth factor subunit A (PDGFA) and its receptor α (PDGFRA) as major drivers of GBM development. Accordingly, several drugs have been developed that target PDGFRA, including imatinib. However, clinical trials failed to show any anti-tumor effects, despite promising pre-clinical studies [208,209,210,211]. Recently, Gai et al. uncovered a potential mechanism by which PDGFA signaling could be mediated, independent of PDGFRA [212]. They observed that EphA2 interacts with PDGFA and upregulates PDGF signaling targets (Figure 4). In fact, the analysis of available GBM patient datasets showed that an upregulation of EphA2 and PDGFRA was associated with poor patient prognosis. In addition, EphA2 expression was associated with the viability and invasiveness of GBM cells. Further, EphA2 was responsible for upregulating genes involved in a malignant GBM phenotype. However, the simultaneous inhibition of PDGFRA and EphA2 suppressed GBM cell growth in vitro and in vivo, thus encouraging clinical studies to be carried out targeting EphA2 protein in GBM treatment (Table 3).

CD147 expression was reported to be closely associated with poor clinical characteristics in glioma patients, including poor survival. Moreover, CD147 expressing glioma showed a higher percentage of 5-year relapse compared with CD147 negative glioma patients [204,213]. CD147 is an extracellular matrix metalloprotease (MMP) inducer that is involved in inducing the production of MMP-1, -2, -3, -9, -14 and -15. These MMPs have been correlated with tumor invasion, metastasis and reduced survival in human glioma [214]. Thus, targeting the CD147-MMP axis could prove to be beneficial in GBM treatment.

CD161 was identified in a scRNA-sequencing analysis of IDH-wildtype and IDH-mutant GBM as a potential inhibitory receptor on TILs [215]. CLEC2D, the ligand for CD161, was found to be expressed by malignant cells and myeloid cells. Further analysis showed that CD161 receptor inhibited T cell function, including cytotoxicity and cytokine secretion. Accordingly, Mathewson et al. genetically inactivated the CD161 gene or blocked the CD161 function using an antibody. This led to an enhancement in the T cell-mediated killing of glioma cells in vitro and in vivo. The study provides a novel ligand-receptor pathway that could be targeted to harness the cytotoxic potential of T cells in glioma therapy.

CD70, identified as a TNF-related cell surface ligand, plays an important role in T, B and NK cell activation via its interaction with the CD27 receptor. Interestingly, CD70 is constitutively overexpressed by IDH wildtype primary and recurrent GBMs and associated with poor patient survival. Moreover, the CD70-CD27 axis has been associated with immune evasion and tumor progression [216]. Recently, it was shown that CD70 expression is associated with the infiltration of T cells, but it also induces the cell death of CD8+ T cells in GBM [217]. Moreover, on targeting CD70 expressing glioma cells using CAR-T cells, tumor regression was observed in glioma mouse models. Although these studies provide another potential target in glioma therapy, it would be important to see whether their effects can be translated into human patients. Other than monotherapies, combinatorial approaches involving multiple therapies are being employed. A new CAR comprising tandemly arranged IL-13 and EphA2 single-change variable fragment (scFv), called TanCAR, selectively killed GBM cells by recognizing the IL-13Rα2 or EphA2 alone or together in vitro. In vivo experiments showed greater tumor regression by TanCAR-redirected T cells compared with single CAR-T cells [218].

Furthermore, NK cells are being engineered with CARs for targeting GBM. Due to the failure of rindopepimut and EGFRvIII-specific vaccines facing multiple challenges, Genßler et al. genetically modified the human NK cell line NK-92 to express CARs identifying epitopes against EGFR, EGFRvIII or those common to both proteins (dual-specific CAR-NK cells). The dual-specific NK cells were observed to be superior in their ability to recognize and lyse both types of tumor cells in vitro compared with mono-specific CAR-NK cells. By introducing dual-specific CAR-NK cells expressing cetuximab-based CAR (cetuximab: EGFR inhibitor), lysis of both EGFRvIII and wildtype EGFR expressing glioma cells in vivo was observed in a GBM mouse model. Moreover, compared with EGFR-vIII specific vaccines, CAR-NK cells reduced the risk of immune escape due to antigen loss [219]. In a similar study, ErbB2-CAR was engineered into the NK-92 cell line, which lysed all ErbB2-positive established and primary GBM cells. In vivo treatment with ErbB2-CAR-NK cells also proved to be beneficial with GBM xenograft mice showing a marked extension of symptom-free survival [220].

Although pre-clinical studies provide impressive results for a treatment regimen, most of them fail to produce an efficacious anti-tumor effect in clinical trials. Commonly known challenges involve tumor heterogeneity, antigen loss and an immunosuppressive environment that allow GBM cells to evade the immune response. Moreover, positive outcomes from pre-clinical tumor models lead to high expectations for successful clinical application. However, although animal models are beneficial for studying tumor development and testing new treatments, these models have not yet been able to fully recapitulate human gliomas. In this context, while GL261 cells have become the standard for glioma development in many mouse models, they are largely immunogenic in nature. Since human GBM is a cold tumor, it has become necessary to use different mouse glioma cell lines that phenocopy human gliomas more closely. Recent reports show that SB28 is a suitable model for optimizing GBM immunotherapy [221,222]. However, it is important to tread with caution, since additional characterization of the cell line needs to be performed along with studying the microenvironment that is induced by these cells. Similarly, monotherapies seem to neglect the ever-changing GME, therefore it would be important to use combinatorial approaches against GBM in the future.

Nonetheless, the pre-clinical studies have shown promise and results for the clinical trials are eagerly awaited. Table 3 provides a list of ongoing clinical trials involving different immunotherapeutic approaches against potential target candidates described in this section.

## 5. Conclusions and Perspectives

The evolving nature of immunotherapies and glioblastoma has revolutionized the field of tumor biology and immune evasion. While conventional therapeutic approaches involving radiotherapy, chemotherapy and surgery have been successful in treating different cancers such as breast cancer, colorectal cancer and lymphomas, the heterogeneous nature of GBM and its inherent immunosuppressive environment have been major challenges in glioma therapy, even with the advent of immunotherapeutic treatments.

An ongoing task is the identification of new candidate targets that can be inhibited or stimulated in the context of immune cell function. In this regard, Shaim et al. reported that targeting the αv integrin/TGF-β axis helps enhance NK cell function against GBM stem cells in vitro [110]. Recently, Karimi et al. found that a subset of neutrophil-like macrophages positive for myeloperoxidase (MPO) could be beneficial for the survival of GBM patients [223]. Thus, inhibiting the αv integrin/TGF-β axis or enhancing the population of MPO+ macrophages, along with employing known immunotherapeutic techniques, needs to be tested as an alternative glioma therapy. The immune landscape, though modulated by the glioma cells, can still be used for our benefit, as discussed in Section 3. In addition to this, newer target candidates described in Section 4 will also play an important role. In fact, with the advent of new technology involving single-cell sequencing and multi-omics, studies have uncovered a plethora of new potential targets for glioma immunotherapy. Among these studies, S100A4 was identified as a highly expressed gene in the Tregs, exhausted T cells and pro-tumorigenic myeloid cells infiltrating human glioma [224]. On deletion of S100A4, T cell activation increased; moreover, an enhancement in myeloid cell phagocytosis was observed. On similar lines, the single-cell characterization of macrophages identified a macrophage receptor with a collagenous structure (MARCO) protein as a pro-tumorigenic marker [225]. The authors found that MARCO was highly expressed in glioblastoma while being absent in low grade gliomas. Furthermore, it was associated with poor patient prognosis. Other than these studies, various other studies focusing on glioma immunotherapy by involving single-cell and multi-omics strategies have been reviewed elsewhere [226,227]. It is important to point out that, while such studies prove to be promising, we need to be cautious about off-target effects. For example, certain off-target effects from CAR-T cells can be suppressed by combining them with vaccine therapy.

Results from ongoing clinical trials (Table 3) are awaited along with the execution of new clinical trials for immunotherapeutic treatments against recently discovered targets. Moreover, studies to understand the failure of certain immunotherapies by identifying the cause and possible ways to counter it in clinical trials are equally important as testing new ones. In conclusion, the ever-evolving nature of the GME necessitates employing immune cells as a double-edged sword by the application of combinatorial immunotherapeutic approaches that trigger the glioma immune landscape to fight against GBM.

## Figures and Tables

**Figure 1 cancers-15-02024-f001:**
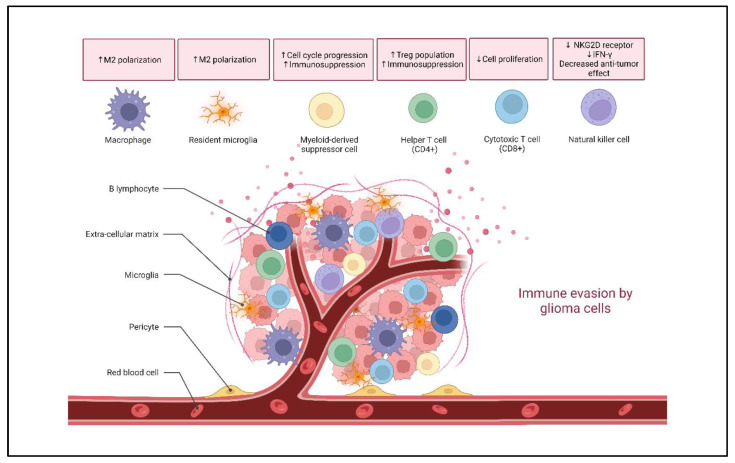
Effect of tumor cells on glioma immune landscape. Glioma cells modulate immune cell functions in the tumor microenvironment, resulting in an immunosuppressive phenotype through secretion of various factors. Therefore, activation of NK and CD8+ T cell is downregulated. Further, the immunosuppressive environment is supported by macrophages polarizing to an anti-inflammatory M2 state and recruitment of MDSCs and Tregs. Created with BioRender.com (accessed on 14 March 2023).

**Figure 2 cancers-15-02024-f002:**
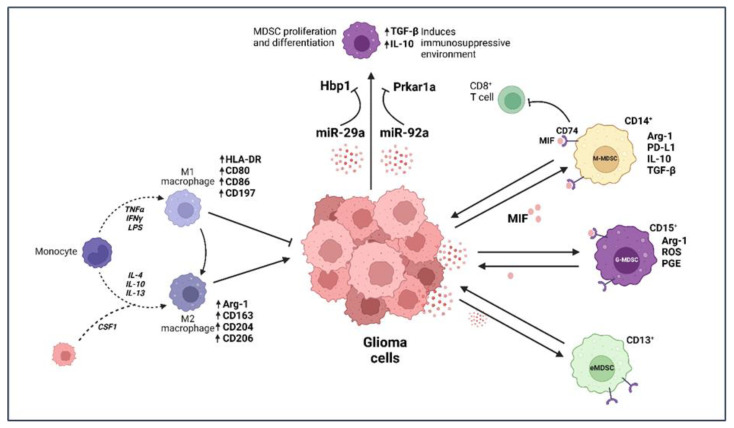
Myeloid immune cells induce immunosuppression within glioma microenvironment. IFNγ—interferon gamma, Hbp1—high-mobility group box transcription factor 1, Prkar1a—protein kinase cAMP-dependent type I regulatory subunit alpha, Arg-1—arginase-1, TGF-β—transforming growth factor β, CSF1—colony stimulating factor 1. Created with BioRender.com (accessed on 14 March 2023).

**Figure 3 cancers-15-02024-f003:**
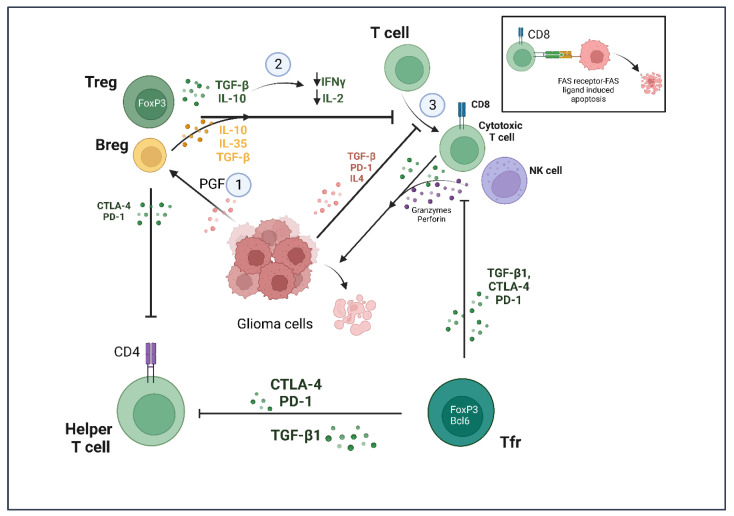
Summary of lymphocyte response against glioma cells. (1) Glioma cells secrete placenta growth factor (PGF), which differentiates infiltrating B cells into regulatory B (Bregs) cells. (2,3) Tregs and Bregs induce an immunosuppressive environment and inhibit development of CD8+ cytotoxic T cell and its cytotoxic effect on tumor cells. Tfr cells inhibit helper and killer T cell function by secretion of PD-1, CTLA-4 and TGF-β1 signaling, while also inhibiting NK cells. NK cells lyse glioma cells by secreting granzymes and perforin. Created with BioRender.com (accessed on 14 March 2023).

**Figure 4 cancers-15-02024-f004:**
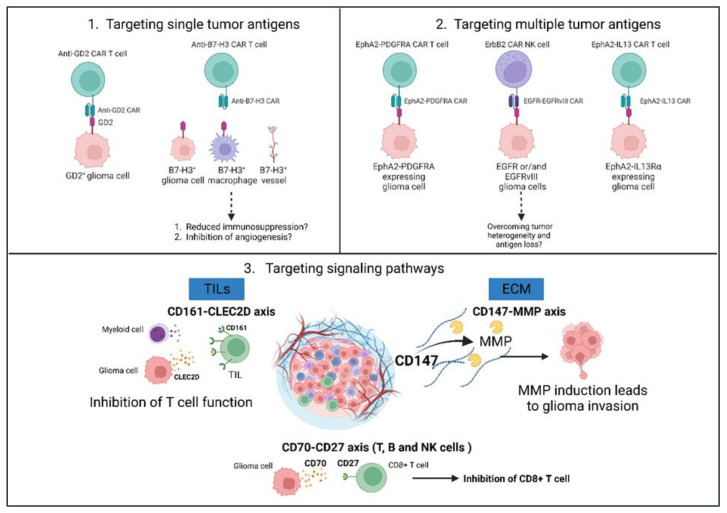
Potential targets for glioma immunotherapy. (1) Single antigens such as GD2 or B7-H3 were targeted using CAR-T therapy in pre-clinical trials and are currently being tested as part of ongoing clinical trials. (2) Targeting multiple antigens is considered an alternative and more efficient strategy compared with targeting a single antigen in glioma therapy. (3) Targeting signaling pathways is also being tested as an alternative route in glioma therapy. GD2—disialoganglioside; B7-H3—B7 homolog 3; EphA2—ephrin receptor A2; ErbB2—erythroblastic leukemia viral oncogene homolog 2; PDGFRA—platelet derived growth factor receptor α; MMP—matrix metalloproteinases; CLEC2D—C-type lectin domain family 2 member D. Created with BioRender.com (accessed on 14 March 2023).

**Table 1 cancers-15-02024-t001:** Model systems used in pre-clinical glioma research.

Immune Cells Involved	Molecule(s) Significant for Glioma Research	Model Systems Used	References
Microglia and macrophages	Glioma derived CSF-1	Mouse models, human GBM tumor spheres, cell lines	[45,46]
	Arg1+ exosomes	Cell culture	[47]
	AEG1	Bioinformatic analysis (TCGA, GTex, CGGA) of human samples, cell lines, co-culture analysis	[48]
MDSCs	NA	GBM patient blood samples + tumor tissue, mouse models	[58]
	MIF	Co-culture assays, GBM patient samples, syngeneic mouse models	[71]
Cytotoxic T cells	PD-1	Metadata analysis of glioma samples from published studies	[78]
Tregs + Tfr cells	PD-1, CTLA-4	Human tumor samples, syngeneic mouse models, tumor cell lines	[79]
Tfr cells	NA	Resected glioma samples from patients	[80]
B lymphocytes	Glioma derived PFG	Primary cell culture	[81]
NK cells	TGF-β and NKG2D	Blood samples from glioma patients	[82]
	IFN-γ	Human GBM tissue samples	[83]

NA—not available.

**Table 2 cancers-15-02024-t002:** Currently available immunotherapies against GBM.

Immunotherapy	Clinical Trial	Immune Response	Reference
**CAR-T therapies**			
IL13Rα2-CAR-T cells	Phase I	Naïve and memory T cells	NCT02208362 NCT04003649 (with immune checkpoint inhibitors)
CMV-specific T cells	Phase I/II	Cytotoxic T cells	NCT02661282
HER2-CAR-CMV-T cells	Phase I	T cells	NCT01109095
**Vaccines**			
Rindopepimut	Phase II Phase III Phase II	EGFRvIII-specific humoral immune response	NCT01498328 [111] NCT01480479 [112] NCT00458601 [113]
IMA950	Phase I/II	CD8+ response and sustained T helper 1 CD4+ T cell response	NCT01920191 [114] NCT01222221
DCs vaccine (PERCELLVAC)	Phase I	Tumor-associated antigen specific CD4+ and CD8+ T cell response	NCT02709616 [115]
CMV pp65 DC vaccine	Phase I Phase II Phase I	Expected activation of CD4+ and CD8+ cells	NCT03615404 NCT02366728 NCT02529072
SurVaxM peptide vaccine	Phase II	Preliminary results do not discuss immune response	NCT02455557
HSPCC-96 vaccine	Phase II	Low PD-L1 expression on myeloid immune cells showed better survival	NCT00905060 [116]
DSP-7888	Phase III Phase I	1. WT-1 specific CTL induction activity not observed in a dose-dependent manner (NCT02498665). 2. Higher WT1-specific CTL induction was observed intradermally than subcutaneously (NCT02498665).	NCT03149003 NCT02498665
AV-GBM1	Phase II	Pro-inflammatory response: increase of Th1, Th2 and Th17 pathway markers as well as B-cells, NK cells and cytotoxic T-lymphocytes	NCT03400917 [117]
**Immune checkpoint inhibitors**			
Nivolumab	Phase II	NA	NCT02550249
Pembrolizumab	Phase II	Infiltration of T cells but CD68+ macrophages predominate (NCT02337686)	NCT02337491 NCT02337686
Ipilimumab	Phase I	NA	NCT02311920 [118] (with Nivolumab)
**Oncolytic viruses**			
PVSRIPO	Phase I	Reduction of Tregs and onset of homeostatic reconstitution of effector T cells	NCT01491893 [119]
DNX-2401 (Tasadenoturev)	Phase I Phase II	CD8+ and T-bet+ cell infiltration (NCT00805376)	NCT01956734 NCT02798406 NCT00805376
G207	Phase I/II Phase I	Short term CD4+ and CD8+ T cell response and a low humoral response (NCT00157703)	NCT00028158 [120] NCT00157703 [121]
Ad-RTS-hIL-12 + Veledimex	Phase I	1. Sustained increase of IFN-γ. 2. Increase in percentage of CD3+CD8+ T cells in peripheral blood. 3. No change in CD3+CD4+ T cells or NK cells (NCT03636477).	NCT02026271 NCT03636477 [122]

NA—not available.

**Table 3 cancers-15-02024-t003:** Ongoing clinical trials for GBM employing immunotherapy.

NCT Number	Phase	Type of Immunotherapy	Status	Glioma Targeted
NCT05474378	I	B7-H3 CAR-T	Recruiting	Recurrent IDH wildtype GBM
NCT04099797	I	C7R-GD2 CAR-T	Recruiting	High grade glioma Diffuse intrinsic pontine glioma
NCT04077866	I II	B7-H3 CAR-T + Temozolomide B7-H3 CAR-T	Recruiting	Recurrent GBM Refractory GBM
NCT02575261	I II	EphA2 CAR-T	Withdrawn	Recurrent GBM Metastatic GBM
NCT04045847	I	CD147 CAR-T	Recruiting	Recurrent GBM
NCT05465954	II	Immune checkpoint inhibitor + IL-7	Recruiting	Recurrent GBM
NCT03360708	I	Tumor lysate-pulsed autologous dendritic cell vaccine	Active, not recruiting	Recurrent GBM
NCT01567202	II	DC vaccine with antigens from glioma stem-like cells (A2B5+)	Recruiting	Newly diagnosed GBM Secondary GBM
NCT05100641	III	DC vaccine (AV-GBM1)	Not yet recruiting	Primary GBM
NCT03661723	II	Immune checkpoint inhibitor + bevacizumab + radiation	Active, not recruiting	Bevacizumab resistant recurrent GBM
NCT04214392	I	Chlorotoxin (EQ)-CD28-CD3zeta-CD19t-expressing CAR T-lymphocytes	Recruiting	MMP2+ recurrent and progressive GBM
NCT03423992	I	CAR-T based on EGFRVIII, IL13Rα2, Her-2, EphA2, CD133, GD2 expression	Recruiting	Malignant glioma

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
