# Peer review of "The Glioma Immune Landscape: A Double-Edged Sword for Treatment Regimens"

_cancers, 2023, doi:10.3390/cancers15072024_

Round 1

Reviewer 1 Report

This review paper discusses the immune landscape of glioblastoma, a deadly form of brain tumors, which currently lacks effective therapies. The authors started by introducing gliomas and its most aggressive form, glioblastoma. Overall, the review paper is well written and is comprehensive.

I think the TME and the immune landscape of GBM will influence both current and future therapeutic endeavors for this disease, including immunotherapy, thus, this review addresses a much-needed clinical indication.

I have a couple of minor comments which I wish the authors to address:

Line 129: The M2 state is further categorized into M2a, M2b, M2c and M2d. Can you further elaborate and explain what these subtypes signify?

Line 665: expand the figure legend or add more information in the text. For example, refer to the figure in the text when the Erb2 CAR-NK cells, or the EphA2-IL13 CAR-T cells are described, or cite the reference in the figure legend.

I am not sure of the difference between tables 1 and 2, as both describe clinical trials involving immunotherapies

Author Response

Reviewer 1

We thank the reviewer for the comments and the opportunity to significantly improve our manuscript. In this revision, we have addressed the concerns of the reviewer thoroughly. All changes in the text are highlighted in grey in the revised manuscript. We also list the changes in this point-by-point response by referring to page and line position. Please find below our responses to the reviewer’s comments.

This review paper discusses the immune landscape of glioblastoma, a deadly form of brain tumors, which currently lacks effective therapies. The authors started by introducing gliomas and its most aggressive form, glioblastoma. Overall, the review paper is well written and is comprehensive.

I think the TME and the immune landscape of GBM will influence both current and future therapeutic endeavors for this disease, including immunotherapy, thus, this review addresses a much-needed clinical indication.

Line 129: The M2 state is further categorized into M2a, M2b, M2c and M2d. Can you further elaborate and explain what these subtypes signify?

Authors’ response: We thank the reviewer for the positive assessment and have elaborated further about the subtypes (page 4, line 135-136). The different subtypes have been established based on their marker expressions and cytokine secretion profiles. We have avoided going deeper into the subtypes of these macrophages since it deviates from the main topic. Nevertheless, an additional reference was provided where the reader can find further information.

Line 665: expand the figure legend or add more information in the text. For example, refer to the figure in the text when the Erb2 CAR-NK cells, or the EphA2-IL13 CAR-T cells are described, or cite the reference in the figure legend.

Authors’ response: We have expanded the figure legend as advised (page 17, line 686-691) and refer to the figure at the suggested position in the text (page 18, line 708).

I am not sure of the difference between tables 1 and 2, as both describe clinical trials involving immunotherapies

Table 2 (previously Table 1) represents completed clinical trials that resulted in available immunotherapies. We added additional information about that point in the main text (page 11, line 408-409). Table 3 (previously Table 2) refers to ongoing clinical trials based on pre-clinical studies (page 18, line 715).

Reviewer 2 Report

This is a well-written review about the glioma landscape.

The authors should define "glioma" in line 38 in more detail mentioning that gliomas cmprose astrocytomas, Glioblastoma, and oligodendrogliomas.

In general the authors should try to further strtify the description following tha aspect what was done in cell cultures, animals and humans. For example PD-L1 was not successful in gliomas as was expected at the beginning (line 272-275)

Author Response

Reviewer 2

We thank the reviewer for the comments and the opportunity to significantly improve our manuscript. In this revision, we have addressed the concerns of the reviewer thoroughly. All changes in the text are highlighted in grey in the revised manuscript. We also list the changes in this point-by-point response by referring to page and line position. Please find below our responses to the reviewer’s comments.

This is a well-written review about the glioma landscape.

The authors should define "glioma" in line 38 in more detail mentioning that gliomas cmprose astrocytomas, Glioblastoma, and oligodendrogliomas.

Authors’ response: We thank the reviewer for the positive response. We added the suggested information about the different types of gliomas and their classification (page 2, line 40-42). A reference from the WHO should provide more details for the interested reader.

In general the authors should try to further strtify the description following tha aspect what was done in cell cultures, animals and humans. For example PD-L1 was not successful in gliomas as was expected at the beginning (line 272-275)

Authors’ response: We have now included a table (Table 1, page 10, line 382) to answer this in a comprehensive manner. The limitations of the models used in various studies have also been discussed briefly in each section.

Reviewer 3 Report

The review demonstrated that the different immune cell populations found in the glioma microenvironment and navigate through the various shortcomings of current immunotherapies for glioma. There are some minor issues to improve.

1. With the application of single-cell technology, there are now many new findings on immune cells in glioma, and the authors are advised to update them.

2. Many tumor analyses are currently presented based on multi-omics data from GBM, some of which may be sensitive to immunotherapy and are recommended to be updated by the authors.

Author Response

Reviewer 3

We thank the reviewer for the comments and the opportunity to significantly improve our manuscript. In this revision, we have addressed the concerns of the reviewer thoroughly. All changes in the text are highlighted in grey in the revised manuscript. We also list the changes in this point-by-point response by referring to page and line position. Please find below our responses to the reviewer’s comments.

The review demonstrated that the different immune cell populations found in the glioma microenvironment and navigate through the various shortcomings of current immunotherapies for glioma. There are some minor issues to improve.

With the application of single-cell technology, there are now many new findings on immune cells in glioma, and the authors are advised to update them.

Authors’ response: We thank the reviewer mentioning this important point. The review has already included some studies that use single-cell technology, such as the Tfr study by Eschweiler et al, 2021 (page 8, line 309-316) or the CD161 receptor study by Mathewson et al, 2021 (page 18, line 724-732). Further, we have included recent studies involving single-cell technology in the conclusion section highlighted in grey (page 20, line 798-810).

Many tumor analyses are currently presented based on multi-omics data from GBM, some of which may be sensitive to immunotherapy and are recommended to be updated by the authors.

Authors’ response: We thank the reviewer for emphasizing this important point. In fact, we did mention a study about MPO+ neutrophil-like macrophages and their role in GBM patient survival by Karimi et al, 2023 in the conclusion (page 20, line 792-794). That group is working on a single-cell spatial immune landscape and utilizing high throughput technologies. Furthermore, now we have also cited the recent review by Kaminska et al, 2021 (page 20, line 810) in the conclusion that provides a comprehensive description of results from multi-omics studies in the microenvironment of malignant gliomas. However, we think that focusing further on more multi-omics data would go beyond the scope of the review.

Reviewer 4 Report

the manuscript is very clear, in particular taking into account the subject (immunotherapy). The review is very well structured and clear, both on the description of cells from the innate immune system present in glioblastoma and their possible roles, as well as on the description of cells from the acquired immune response. The authors are very careful introducing the concepts and the pros and cons of each point of view described in the literature.

The description of the possibilities of using CAR-T cells, tumor vaccines or immunocheckpoint inhibitors are very clear. 

 I would appreciate if the authors elaborate a bit more in the legends for figures 1 and 3.

As a very minor comment, I believe that granulolysin can be taken out from the text, it is clear by now that granulolysin is the same as perforin.

Author Response

Reviewer 4

We thank the reviewer for the comments and the opportunity to improve our manuscript. In this revision, we have addressed the concerns of the reviewer thoroughly. All changes in the text are highlighted in grey in the revised manuscript. We also list the changes in this point-by-point response by referring to page and line position. Please find below our responses to the reviewer’s comments.

The manuscript is very clear, in particular taking into account the subject (immunotherapy). The review is very well structured and clear, both on the description of cells from the innate immune system present in glioblastoma and their possible roles, as well as on the description of cells from the acquired immune response. The authors are very careful introducing the concepts and the pros and cons of each point of view described in the literature.

The description of the possibilities of using CAR-T cells, tumor vaccines or immunocheckpoint inhibitors are very clear. 

I would appreciate if the authors elaborate a bit more in the legends for figures 1 and 3.

Authors’ response: We thank the reviewer for the positive assessment and have provided further description in the legends for figures 1 and 3 as suggested (page 3, line 98-101 and page 8, line 302-304, respectively).

As a very minor comment, I believe that granulolysin can be taken out from the text, it is clear by now that granulolysin is the same as perforin.

Authors’ response: We thank the reviewer for bringing up this point. We actually mention the protein granulysin in our text, and not granulolysin. However, to our knowledge granulysin and perforin are not the same. As stated by Crespo et al., 2020 (https://doi.org/10.1016/j.cell.2020.07.019), the cytolytic effectors perforin, granulysin and granzymes have different functions in immune defense. Granulysin, an antimicrobial peptide, selectively disrupts bacterial, fungal, and parasite membranes (Dotiwala and Lieberman, 2019). Perforin is another cytotoxic granule pore-forming protein that selectively injures mammalian but not microbial membranes. Perforin enables the transport of granzymes and granulysin into an infected host cell where granzymes kill the host cell. Within an infected host cell granulysin delivers then the granzymes into intracellular microbes to trigger microptosis (Dotiwala et al., 2016, 2017; Walch et al., 2014). Thus, destroying intracellular microbes by killer cells requires all three cytotoxic effector molecules.